# Transcriptome Response of Differentiating Muscle Satellite Cells to Thermal Challenge in Commercial Turkey

**DOI:** 10.3390/genes13101857

**Published:** 2022-10-14

**Authors:** Kent M. Reed, Kristelle M. Mendoza, Jiahui Xu, Gale M. Strasburg, Sandra G. Velleman

**Affiliations:** 1Department of Veterinary and Biomedical Sciences, University of Minnesota, St. Paul, MN 55108, USA; 2Department of Animal Sciences, The Ohio State University/Ohio Agricultural Research and Development Center, Wooster, OH 44691, USA; 3Department of Food Science and Human Nutrition, Michigan State University, East Lansing, MI 48824, USA

**Keywords:** satellite cell, skeletal muscle, growth selection, turkey, differentiation, hypertrophy

## Abstract

Early muscle development involves the proliferation and differentiation of stem cells (satellite cells, SCs) in the mesoderm to form multinucleated myotubes that mature into muscle fibers and fiber bundles. Proliferation of SCs increases the number of cells available for muscle formation while simultaneously maintaining a population of cells for future response. Differentiation dramatically changes properties of the SCs and environmental stressors can have long lasting effects on muscle growth and physiology. This study was designed to characterize transcriptional changes induced in turkey SCs undergoing differentiation under thermal challenge. Satellite cells from the *pectoralis major* (*p. major*) muscle of 1-wk old commercial fast-growing birds (Nicholas turkey, NCT) and from a slower-growing research line (Randombred Control Line 2, RBC2) were proliferated for 72 h at 38 °C and then differentiated for 48 h at 33 °C (cold), 43 °C (hot) or 38 °C (control). Gene expression among thermal treatments and between turkey lines was examined by RNAseq to detect significant differentially expressed genes (DEGs). Cold treatment resulted in significant gene expression changes in the SCs from both turkey lines, with the primary effect being down regulation of the DEGs with overrepresentation of genes involved in regulation of skeletal muscle tissue regeneration and sarcomere organization. Heat stress increased expression of genes reported to regulate myoblast differentiation and survival and to promote cell adhesion particularly in the NCT line. Results suggest that growth selection in turkeys has altered the developmental potential of SCs in commercial birds to increase hypertrophic potential of the *p. major* muscle and sarcomere assembly. The biology of SCs may account for the distinctly different outcomes in response to thermal challenge on breast muscle growth, development, and structure of the turkey.

## 1. Introduction

Early muscle development is a multifaceted process that involves the proliferation of undifferentiated stem cells in the mesoderm (myoblasts) followed by differentiation and fusion to form multinucleated myotubes. Both early and subsequent muscle growth and repair is dependent upon a population of muscle stem cells (satellite cells, SCs). In the mammalian neonate, SCs comprise ~30% of muscle micronuclei but this number declines to only 1–5% in adult skeletal muscle [1,2]. Based on study of chickens, a similar progression of SCs is thought to exist in birds [3,4]. Satellite cells are also a heterogeneous population, with activation, proliferation and differentiation potentials that change with age [5]. For example, in turkeys maximum SC activity occurs immediately after hatch; proliferation declines after wk 4 and differentiation significantly decreases as early as wk 4 post hatch. Cell culture studies suggest that SCs are also multi-potential as they can be induced to follow osteogenic or adipogenic cellular pathways in addition to myogenesis [6,7,8].

Satellite cells are modulated by the cellular microenvironment and their active state is defined by antagonistic Notch and Wnt signaling, with Notch maintaining expression of *PAX7* (proliferation modulator, Paired box protein Pax-7) and Wnt signaling driving expression of *MYOD* (myoblast determination protein 1) [9,10,11]. Thus the commitment of the activated SCs to myogenesis is marked by *MYOD* expression, downregulation of *PAX7* and an increase in myogenin required for differentiation [12]. Studies in mice have also demonstrated a role of the JAK1/STAT1/STAT3 pathway in both myoblast proliferation and in limiting premature differentiation and a novel role is hypothesized for the JAK2/STAT2/STAT3 pathway in promoting differentiation by regulating signaling molecules in myogenic differentiation [13].

In the early post-hatch period, when avian SCs are highly active mitotically [14,15], perturbation by environmental stimuli can have long lasting effects on muscle growth and physiology [16,17]. Hatchlings have immature thermoregulatory systems and are highly susceptible to the effects of ambient temperature fluctuation [18,19,20] and are thus particularly vulnerable to the adverse effects of climate change. Turkey SCs are thermally sensitive, and expression changes in key myogenic regulatory factors are observed with alterations in temperature [21]. Previous studies by our group have examined differential response of SCs from 7-wk old F-line birds (selected for only 16 wk body-weight) and the Randombred Control Line 2 (RBC2) from which the F-line was selected [22,23]. In these select research lines, differentiation of SCs was directly influenced by temperature with greater effects seen in birds selected for fast growth [24] and with fewer gene expression differences observed in the differentiating cells than proliferating cells [25]. We thus hypothesized that the SCs of faster-growing birds are more sensitive to thermal stress primarily at early points in SC activation.

Recently [26], we expanded these studies by examining thermal stress in proliferating SCs isolated from 1-wk old birds (the period of maximal mitotic activity). This study contrasted the thermal response of proliferating commercial bird SCs with those from the research lines (RBC2 and F-Line) used in previous studies. Thermal challenge during proliferation resulted in significant gene expression differences particularly in pathways involved in signaling and regulation of Ca^2+^ homeostasis, consistent with downstream altered cellular differentiation. The present study was designed to characterize gene expression in differentiating SCs from 1-wk old commercial birds by examining RNAseq profiles following thermal stress. We contrasted the results from SCs of commercial birds with our previous studies of growth-selected research lines. Results identify key genes and pathways affected by thermal challenge, identifying gene targets for future investigations.

## 2. Materials and Methods

### 2.1. Turkey Myogenic Satellite Cells

The skeletal muscle SCs used in this study were previously isolated according to the protocol of Velleman et al. [27] from the *p. major* muscle of 20 1-wk old males from each of two turkey lines; the RBC2 and commercial turkeys obtained from Nicholas Turkeys, NCT (Lewisburg, WV, USA). The NCT turkeys are commercial meat-type turkeys selected for increased growth rate and breast-muscle yield and the RBC2 turkeys were initiated in 1966 by crossing two commercial large white turkey strains and represent the commercial turkeys at the time [28]. Only males are represented in the cell lines to avoid any sex effects.

The turkey SCs were replicate plated and proliferated as described in Reed et al. [26]. After 72 h of proliferation at 38 °C, the growth medium was replaced with a reduced-serum differentiation medium containing DMEM, 3% horse serum (Gemini BioProducts, Sacramento, CA, USA), 1% antibiotics-antimycotics (Gemini BioProducts), 0.1% gentamicin (Gemini BioProducts), and 1 mg/mL bovine serum albumin (BSA, Sigma Aldrich, St. Louis, MO, USA). Cells were cultured in 95% air/5% CO_2_ incubators at 38 °C (control) or at an experimental temperature (33 °C or 43 °C) and medium changed every 24 h. The control temperature of 38 °C is approximately that measured in newly hatched poults (38.0–38.5 °C, Strasburg, unpublished data) and was the temperature used during primary generation of the SCs to the 4th passaged generation. At the conclusion of the 48 h treatment, cell medium was removed, cells were rinsed with PBS, collected into RNAzol RT (Sigma Aldrich) and held at −80 °C until RNA isolation. 

### 2.2. RNA Isolation and Sequencing

Total RNA extraction, sample QC, library preparation were as described in Reed et al. [26]. Briefly, Total RNA was isolated by RNAzol RT extraction, DNase-treated (Turbo DNA-freeTM Kit, Ambion, Inc., Austin, TX, USA), and stored at −80 °C. Concentration and quality of RNAs were assessed by spectrophotometry (Nanodrop 1000). Each sample was then quantified by RiboGreen Assay (Invitrogen Corp., Waltham, MA, USA) and RNA integrity confirmed on the 2100 Bioanalyzer (Agilent Technologies, Santa Clara, CA, USA). For each treatment group, replicate samples were sequenced (*n* = 2 replicates per treatment group). Library preparation and sequencing was conducted at the University of Minnesota Genomics Center on the NovaSeq SP platform using v1.5 chemistry (Illumina, Inc., San Diego, CA, USA) to produce 51-bp paired-end reads.

### 2.3. RNAseq Data Analyses

Sequence processing and mapping were as described in Reed et al. [26] using the turkey genome (UMD 5.1, ENSEMBL Annotation 104). Principal component analysis (PCA) was performed in CLC Genomics Workbench (CLCGWB v. 10.1, Qiagen, Redwood City, CA, USA) on normalized read counts. Gene IDs were determined as described in Reed et al. [26]. Differential gene expression and ANOVA (Bonferroni and False Discovery Rate [FDR] corrected) was performed in CLCGWB using standard work flow and with the RBC2 cells set as control for between-line comparisons and the 38 °C treatment groups set as control for temperature/treatment comparisons (Wald test). Pair-wise comparisons between treatment groups were made in the Bioconductor (3.2) R package DESeq2 [29]. For all comparisons, *p*-values less than 0.05 were considered significant. Affected genes and pathways were investigated with Ingenuity Pathway Analysis [30], Gene Ontology (GO analysis) and PANTHER Overrepresentation Tests (GO Ontology database doi:10.5281/zenodo.6799722 Released 1 July 2022, [31]) http://geneontology.org/ (accessed 8 October 2022). 

## 3. Results

Total RNA was used to construct individual barcoded libraries for the SC cultures (2 lines × 3 treatments × 2 replicates = 12 total). Sequencing produced over 273M paired-end reads (Table 1) accessioned in the NCBI’s Gene Expression Omnibus (GEO) repository as part of SRA BioProject PRJNA842679. The number of reads per library ranged from 19.2 M to 25.9 M (average 22.8 M). Read quality was consistently high and Q values ranged from 35.8 to 36.3. Replicate libraries produced comparable results with an average difference between replicates of 2.9 M reads.

### 3.1. Gene Expression

Evidence for expression (at least one mapped read in at least one treatment group) was observed for 15,771 annotated genes representing 87.8% of the ENSEMBL turkey gene set (~90% of protein-coding genes) (Appendix A). The number of observed genes per library ranged from 13,537 to 14,000 (average 13,752.0), and the number of observed genes per treatment ranged from 13,653 to 13,949 (~80% of gene set) (Table 1). Distribution of expressed protein-coding genes (shared and unique) in treatment group comparisons is summarized in Table 2. The total number of shared genes (expressed in all treatments) was 12,487 with the number of unique genes ranging from 367 to 651 in within-group comparisons. Uniquely expressed genes were higher in the RBC2 cells at 33 °C and 38 °C, but greater in the NCT SCs at 43 °C. 

Normalized read counts were used to conduct a principal component analysis (PCA) to partition variation among treatment groups (Figure 1). Within the first two principal components, treatment groups clustered distinctly irrespective of line and replicate treatment pairs occurred as nearest neighbors within the PCA space.

Several genes that serve as markers of the muscle differentiation cascade were included in the RNAseq data. Contrasting expression in differentiating cells with that of proliferating cells [26] found significant up regulation for *MYOD1*, *PAX7*, *MYOG* (myogenin), *DLL1* (delta like canonical Notch ligand 1), and *MSTN* (myostatin) (Table 3). Significantly down-regulated genes included *MYF6* (myogenic factor 6), *PPARγ* (peroxisome proliferator activated receptor γ), and *LIF* (LIF interleukin 6 family cytokine).

### 3.2. Differential Expression

Seven two-way gene expression contrasts were made based on temperature treatment (cold and hot) and turkey line (RBC2 and NCT). Temperature effects were examined in four pairwise, within-line comparisons: cold-versus control (33 °C vs. 38 °C) and hot-versus control (43 °C vs. 38 °C) (Appendix A). On average, more genes were significantly affected (FDR *p*-value < 0.05) by heat (43 °C) treatment than by cold (33 °C) although a greater number of DEGs (|Log_2_FC| > 2.0) were seen in the NCT cells with cold treatment and a greater number in the RBC2 cells with heat. A large portion of DEGs identified in the treatment comparisons (36–56%) were unique to treatment groups (temperature/line) (Appendix A).

#### 3.2.1. Effect of Cold Treatment

A total of 575 DEGs were identified in the 33 °C vs. 38 °C comparisons (Figure 2, Appendix A). A total of 125 DEGs (21.7%) was common to the two genetic lines and the number of unique DEGs was higher in the NCT cells compared to the RBC2 (331 [57.6%] and 119 [20.7%], respectively). In both lines, the number of down regulated DEGs was greater than those up regulated.

In the RBC2 SCs 244 DEGs were identified (|Log_2_FC| > 2.0) with 181 (74.2%) being down regulated by cold treatment. The 40 DEGs with the greatest expression change in each comparison are presented in Figure 3. DEGs with the greatest down regulation included *ENSMGAG00000018824* (a novel protein coding gene with BLAST homology to Proline-Rich Protein 33, *PRR33*), *ENSMGAG00000001884* (a myosin heavy chain homolog), *ADCY8* (adenylate cyclase 8), and *TRABD2B* (TraB domain containing 2B). In humans, PRR33 is predicted to act within the response to wounding, ADCY8 catalyzes the formation of cyclic AMP from ATP [32], and TRABD2B is involved in several processes, including negative regulation of Wnt signaling pathway [33]. DEGs with the greatest up regulation included *IGSF21* (immunoglobin superfamily member 21), *SOX10* (SRY-box transcription factor 10), and *LOC100545586*, (microsomal triglyceride transfer protein large subunit-like). As a transcription factor SOX10 is thought to be involved in the regulation of embryonic development and in the determination of the cell fate [34]. Microsomal triglyceride transfer proteins are involved in lipoprotein assembly [35].

Overrepresentation tests (PANTHER) of the DEGs in the RBC2 SCs (91.7% of IDs mapped to *Gallus gallus* gene set) found greatest enrichment for the GO Biological Processes of *Sarcomere organization* (40.82-fold, *p* = 7.97 × 10^−07^), *Myofibril assembly* (24.35x, *p* = 2.85 × 10^−07^), and *Muscle cell development* (23.81x, *p* = 3.46 × 10^−07^). Greatest enrichment for the GO Molecular Function category was observed for *Troponin I binding* (>100-fold, *p* = 9.88 × 10^−04^), *Troponin C binding* (>100x, *p* = 9.88 × 10^−04^) and *Tropomyosin binding* (45.92x, *p* = 4.10 × 10^−05^). Cellular Component enrichment was greatest for *Troponin complex* (48.71-fold, *p* = 3.86 × 10^−05^), *Myofilament* (39.48, *p* = 3.01 × 10^−06^), and *Striated muscle thin filament* (39.48, *p* = 3.01 × 10^−06^). 

Ingenuity pathway analysis (IPA, [30]) was used to infer biological functions in SCs significantly affected by cold treatment. Pathway Activity Analysis which compares the observed expression of genes relative to that expected under pathway activation found the canonical pathways with the largest z-scores (|z|) were negative and included *Oxidative phosphorylation* (z-score = −5.1, −log(pval) = 3.23), *Superpathway of cholesterol biosynthesis* (−3.0, 2.84) and *Gluconeogenesis I* (−3.0, 2.49). 

In the NCT SCs, 456 DEGs were identified (|Log_2_FC| > 2.0) with 313 (68.6%) being down regulated by cold treatment (Figure 2). DEGs with the greatest down regulation included *ENSMGAG00000018824* and *ENSMGAG00000001884* shared with the RBC2 SCs (Figure 3, Appendix A). Down regulated DEGs included myosin heavy chain genes *LOC100544198*, *LOC100544198*, and *MYH1E* (average Log_2_FC = −6.8) and the myosin light chain gene *MYL3* (Log_2_FC = −6.64). *MYL3* was also significantly down regulated in the RBC2 SCs as was stathmin 4 (*STMN4*), a gene thought to enable tubulin binding activity. Highly up-regulated genes include two (*ADAMTS3* and *ENSMGAG00000021208*) with predicted function in collagen formation (Figure 3). ADAMTS3 is a protease that may play a role in the processing of type II fibrillar collagen [36]. *ENSMGAG00000021208*, annotated as a novel protein coding has BLAST similarity to translocation associated membrane protein 2 (*TRAM2*). TRAM2 is predicted to couple the activity of the ER Ca^2+^ pump SERCA2B to increase local Ca^2+^ concentration at the site of collagen synthesis [37].

Analysis of the DEGs with |Log_2_FC| > 2.0 (283 IDs mapped to *G. gallus* gene set) in PANTHER found greatest enrichment in the GO Biological Process category for *Sarcomere organization* (22.04-fold, *p* = 9.68 × 10^−05^) and *Myofibril assembly* (17.09x, *p* = 4.65 × 10^−08^) and *Striated muscle cell development* (16.71x, *p* = 5.90 × 10^−08^). When all significant genes with (FDR *p* < 0.05) were included (3954 mapped IDs), *Translation* (1.78x, *p* = 9.42 × 10^−04^) and *Peptide biosynthetic process* (1.70x, *p* = 6.75 × 10^−03^) showed greatest gene enrichment in the GO Biological Process category. Analysis of DEGs in IPA found the canonical pathways with greatest z-score activation (|z|) were *Oxidative phosphorylation* (z = −7.2, −log(pval) = 11.0), *NF-κB signaling* (3.78, −log(pval) = 0) and *FLT3 Signaling in hematopoietic progenitor cells* (3.65, −log(pval) = 6.6). FLT3L/FLT3 signaling controls many cellular processes including the normal development of stem cells [38]. 

In the comparison of cold treated cells, 125 DEGs were shared between the RBC2 and NCT SCs. Of these all responded similarly with 113 being down regulated and 12 up regulated by heat treatment in both lines (Appendix A). Comparison analysis of the cold-treated RBC2 and NCT DEGs in IPA identified *Oxidative phosphorylation* as the canonical pathway with the greatest average |z|-score (−7.21 and −5.10 in NCT and RBC2 cells, respectively, Figure 4). Directional activation was generally consistent between lines, except for *Cholesterol biosynthesis*, where positive z-scores occurred in NCT and negative scores in RBC2.

#### 3.2.2. Effect of Heat Treatment

A greater number of DEGs were identified in the 43 °C vs. 38 °C comparisons (Figure 2, Appendix A) with fewer DEGs (115, 14.5%) shared between lines than in the 33 °C vs. 38 °C comparison. The number of unique DE genes was higher in the RBC2 cells compared to the NCT SCs (410 [50.3%] versus 280 [34.8%]) and 115 (14.3%) DEGs were common to the two genetic lines. In contrast to the cold treated cells, the number of DEGs up regulated by heat was greater in both lines than those down regulated.

In the RBC2 SCs, 525 DEGs were identified (|Log_2_FC| > 2.0) with 328 (62.5%) being up regulated. The 40 significant DE genes with the greatest expression change in each comparison are presented in Figure 5. DEGs with the greatest down regulation in the RBC2 SCs included *ND4L* (NADH dehydrogenase subunit 4 L), and G-coupled receptors *ADRA1D* (adrenoceptor α 1D), *NTSR1* (neurotensin receptor 1) and *ADGRL4*. These receptors can activate mitogenic responses. For example, ADRA1D has been implicated in growth-promoting pathways [39].

DEGs with the greatest up regulation included several myosin associated genes: *MYH1E* (myosin heavy chain 1E), *MYBPC1* (myosin binding protein C1), *MYH2* (myosin heavy chain 2), *ENSMGAG00000001884* (myosin motor domain-containing protein), and *LOC100544198* (myosin heavy chain) suggesting promotion of muscle development by heat stress. Within the DEGs, significant enrichment was observed for the GO Biological process of *Muscle contraction* (5.97-fold, *p* = 2.57 × 10^−02^), supporting this observation. Analysis of DEGs in IPA found the canonical pathways with greatest activity z-scores (|z|) were *Oxidative phosphorylation* (z-score = 5.1, −log(pval) = 4.5), *Cell cycle control of chromosomal replication* (z-score = −3.5, −log(pval) = 3.52) and *FLT3 Signaling in hematopoietic progenitor cells* (z-score = −3.087, −log(pval) = 6.5).

In the NCT SCs, 395 DEGs were identified (|Log_2_FC| > 2.0) with 248 (62.8%) being up regulated by heat treatment (Figure 2, Appendix A). DEGs with the greatest down regulation included *TMED6* (transmembrane p24 trafficking protein 6), *AKR1D1* (aldo-keto reductase family 1 member D1) and *ENSMGAG00000020839* (novel protein with BLAST similarity to collagen α-2(I) chain-like, *LOC104913712*). Two of the 20 most down-regulated DEGs (*ENSMGAG00000019963* and *ENSMGAG00000022949*) were shared with the DEGs found in the RBC2 SCs. Both of these are annotated in ENSEMBL as novel protein coding genes. *ENSMGAG00000019963* has BLAST similarity to uncharacterized protein C9orf152 and *ENSMGAG00000022949* is likely a ncRNA. Highly up-regulated genes included *LOC100546217* (protein TENP, also known as *BPIFB2* BPI fold containing family B member 2), *TACR1* (tachykinin receptor 1), and *ENSMGAG00000020551* (annotated novel protein with BLAST similarity to *CLASP2*, CLIP-associating protein 2). BPIFB2 is a lipid transfer/lipopolysaccharide binding protein [40]. The tachykinin receptor 1 (TACR1) is characterized by interactions with G proteins [41], and this DEG was also significantly up regulated in the heat-treated RBC2 SCs. CLASP2 is a tracking protein that promotes the stabilization of dynamic microtubules [42]. 

Analysis of the DEGs with |Log_2_FC| > 2.0 (205 IDs mapped to *G. gallus* gene set) in PANTHER found greatest enrichment in the GO Biological Process category *Cell adhesion* (4.00-fold, *p* = 1.00 × 10^−05^). Based on IPA analysis, the canonical pathway of *Oxidative phosphorylation* was significantly affected in the NCT SCs as it was for the RBC2 cells (Figure 6). However, in contrast to the RBC2 cells, the activation z-score was negative (z-score = −6.6, −log(pval) = 10.6). Significant positive z scores were seen for the *Pulmonary fibrosis idiopathic signaling pathway* (5.9) and *Cardiac hypertrophy signaling* (5.6).

In the comparison of heat-treated cells, 115 DEGs were shared between the RBC2 and NCT SCs. Of these, 61 were up regulated by heat treatment in both lines, 39 were down regulated and 15 responded directionally opposite (Appendix A). Interestingly, 13 of these 15 DEGs were up regulated in the NCT cells with only two showing higher expression in the RBC2 SCs. The two DEGs showing exclusive up regulation in the RBC2 SCs were *LOC104916271* (osteocalcin-like, Log_2_FC = 2.1317 and −5.1310 in the RBC2 and NCT SCs, respectively) and *RHCG* (Rh family C glycoprotein, Log_2_FC = 2.4501 and −2.1274, in the RBC2 and NCT SCs, respectively). Osteocalcin is thought to play a role in metabolic regulation and is secreted solely by osteoblasts [43]. 

The 13 DEGs showing higher expression in the NCT cells included the transporter *ABCC1* (ATP binding cassette subfamily C member 1), membrane proteins *CLTC* (clathrin heavy chain) and *FREM1* (*FRAS1* related extracellular matrix 1), *INF2* (Inverted formin, FH2 and WH2 domain containing), *LOC100550636* (C2 domain-containing protein 3), *LOC104910681* (tyrosine-protein phosphatase non-receptor type 13-like), *TESK2* (testis-specific kinase 2) and 6 uncharacterized/novel proteins (*ENSMGAG00000015050*, *ENSMGAG00000018377*, *ENSMGAG00000019695*, *ENSMGAG00000020631*, *ENSMGAG00000021377*, and *ENSMGAG00000022243*). The different responses of the NCT and RBC2 SCs to the heat treatment in several canonical pathways is evident in the IPA Comparison analysis illustrated (Figure 6). Directional differences in the activation score were seen for the majority of the 20 pathways with the highest composite z-scores.

### 3.3. Effects of Selection

#### 3.3.1. Line Differences: NCT versus RBC2 SCs

The number of DEGs was lower in the between-line contrasts than the within-temperature contrasts (Table 2, Figure 7). A greater number of genes were down regulated in the NCT SCs compared to the RBC2 SCs at 33 °C and 38 °C, but more DEGs were up regulated at 43 °C (Figure 7). Common to all three temperature contrasts were 26 DEGs, that with the exception of *ETNPPL* (ethanolamine-phosphate phospho-lyase, also known as *AGXT2L1*), were all were down regulated in the NCT SCs compared to RBC2 (average Log_2_FC = 2.6). Included in the 25 down-regulated DEGs were *BMP3* (bone morphogenetic protein 3), *DMP1* (dentin matrix acidic phosphoprotein 1), *ENSMGAG00000022282* (ubiquitin-conjugating enzyme E2 R2-like), *KCMF1* (E3 ubiquitin-protein ligase), *GOLPH3* (Golgi phosphoprotein 3), *NTRK2* (neurotrophic receptor tyrosine kinase 2), *LOC100544159* (nipped-B-like protein), *LOC100545914* (ATP synthase subunit α), *LOC100550847* (spindlin-W-like), *LOC100544169* (thioredoxin-like protein 1), *LOC100548376* (transitional endoplasmic reticulum ATPase), *LOC100303669* (ubiquitin-associated protein 2), *LOC104915564* (zinc finger RNA-binding protein-like), *LOC104914913* (zinc finger SWIM domain-containing protein 6-like), *ROBO2* (roundabout guidance receptor 2), and 10 novel/uncharacterized loci. Among these DEGs, the membrane-bound receptor *NTRK2*, is of particular interest in that signaling through this kinase leads to cell differentiation [44]. The ligand BMP3 has been shown to induce bone and cartilage development [45].

At the control temperature (38 °C), 278 DEGs (|Log_2_FC| > 2.0) were identified in the NCT/RBC2 comparison with the majority (52.2%) being down regulated in NCT SCs. Greatest down regulation was seen for *KCMF1* (Log_2_FC = −9.86), *ENSMGAG00000022282* (Log_2_FC = −8.61), and *LOC100544159* (Log_2_FC = −8.34), three loci among the 26 shared with all three temperature contrasts. Greatest up regulation was seen for *MYH1E* (myosin, heavy chain 1E, Log_2_FC = 6.46), *LOC100541145* (vasotocin-neurophysin VT, Log_2_FC = 6.02, homologous to the mammalian arginine vasopressin), and the hydrolase *MINDY4B* (MINDY family member 4B, Log_2_FC= 5.8) likely involved in deubiquitination. 

Overrepresentation tests in PANTHER based on all DEGs (FDR *p* < 0.05) found significant altered expression of genes involved in muscle organization and development (Table 4). When the analysis was limited to DEGs up regulated in the NCT SCs relative to the RBC2 cells, the significant GO Biological Processes of *Mitochondrial ATP synthesis coupled electron transport* (5.64-fold, *p* = 3.53 × 10^−04^), *ATP synthesis coupled electron transport* (5.59x, *p* = 1.70 × 10^−04^), and *Aerobic electron transport chain* (5.56x, *p* = 9.50 × 10^−04^) had the greatest fold change. The GO Biological processes of *Sterol biosynthetic process* (5.57-fold, *p* = 4.23 × 10^−02^) and *Regulation of protein-containing complex disassembly* (3.34x, *p* = 1.55 × 10^−02^) had the greatest fold change when the analysis was limited to only down-regulated DEGs.

The canonical pathway *Oxidative phosphorylation* had the highest activation z-score in IPA (z = 5.39, −log(pval) = 6.25) while the lowest activation score (z = −4.02, −log(pval) = 5.36) was observed for *Cell cycle control of chromosome replication* (Figure 8a). With the exception of *Oxidative phosphorylation*, the general trend in the pathways with the highest activation scores (|z|) was down regulation of associated genes. In the case of the *Oxidative phosphorylation* pathway, associated genes are up regulated in the NCT cells, with the exception of *ND4L* with significantly lower expression (Log_2_FC = −8.23).

In contrast to the control temperature, fewer DEGs were identified between the lines in the cold treatment. At 33 °C, 152 genes showed significant FDR *p*-values with |Log_2_FC| > 2.0 in the NCT SCs compared to the RBC2 cells (Figure 7). The majority of these (75.6%) were down regulated in the NCT SCs compared to the RBC2 cells and 86 DEGs were unique to the 33 °C temperature comparison. Greatest up regulation in the NCT SCs was seen for *LOC100538380*, (protein Wnt-11b-like, Log_2_FC = 5.87), *CHRNB3* (cholinergic receptor nicotinic β 3 subunit, Log_2_FC = 5.54), *SLC13A4* (solute carrier family 13 member 4, Log_2_FC = 5.39), and *ANGPT1* (Angiopoietin-1, Log_2_FC = 5.25) (Appendix A). Members of the Wnt gene family are involved in several developmental processes, including cell proliferation, differentiation, migration, and patterning during development [46]. In humans, angiopoietin 1 is an important mediator in migration, growth, and differentiation of endothelial cells during angiogenesis [47]. Greatest down regulation was seen for *KCMF1* (Log_2_FC = −10.73), *LOC100550847* (Log_2_FC = −10.01), and *LOC100303669* (ubiquitin-associated protein 2, Log_2_FC = −10.17). LOC100303669, involved in the ubiquitination pathway, is unique in that it was not among the 26 DEGs shared with all three temperature contrasts.

Overrepresentation tests in PANTHER based on all DEGs (FDR *p* < 0.05) in the NCT/RBC2 line contrast at 33 °C predicted greatest fold change in the GO Biological Process categories *Exocrine system development* (GO:0035272) and *Sarcomere organization* (GO:0045214) (Table 4). When the analysis was limited to DEGs up regulated in the NCT SCs relative to the RBC2 cells, the GO Biological Process categories of *Regulation of muscle contraction* (4.36-fold, *p* = 3.04 × 10^−02^), *Aerobic respiration* (3.53x, *p* = 1.20 × 10^−02^) and *Striated muscle cell differentiation* (3.4x, *p* = 7.60 × 10^−03^) had the greatest fold change. The GO Biological Process categories of *Mitotic sister chromatid segregation* (5.68-fold, *p* = 2.95 × 10^−07^) and *Sister chromatid segregation* (5.12x, *p* = 1.86 × 10^−07^) had the greatest fold change based on the down-regulated genes. Similar to the control temp analysis, the *Oxidative phosphorylation* pathway had the highest activation z-score in IPA (z = 4.26, -log(pval) = 3.09) while the lowest activation score was observed for *Cell cycle control of chromosome replication* (z = −2.89, −log(pval) = 2.21) (Figure 8b). Differential expression of genes in the pathways with greatest |z|-score was more balanced than at 38 °C with similar percentages of up- and down-regulated genes.

A greater number of DEGs were identified between the lines in the heat treatment than the cold. At 43 °C, 249 genes showed |Log_2_FC| > 2.0 in the NCT SCs compared to the RBC2 cells (Figure 7). In this comparison, the majority of DEGs (55.8%) were up regulated in the NCT SCs compared to the RBC2 cells and 183 were unique to the comparison. Greatest up regulation in the NCT SCs was seen for *FREM1* (FRAS1 related extracellular matrix 1, Log_2_FC= 8.22) and *ADRA1D* (adrenoceptor α 1D, Log_2_FC = 6.19) (Appendix A). In humans, FREM1 is described as an extracellular matrix protein required for epidermal adhesion during embryonic development [48]. The α-adrenergic receptor ADRA1D, helps to regulate growth and proliferation through the influx of extracellular calcium [49]. Greatest down regulation was seen for *KCMF1* (RING-type E3 ubiquitin transferase, Log_2_FC = −10.73), and *LOC100544169* (*TXNL1*, thioredoxin-like protein 1, Log_2_FC = −9.34). Thrioredoxin enables disulfide oxidoreductase activity and reacts to regulate enzyme response to excessive reactive oxygen species [50].

Based on overrepresentation test of all DEGs (FDR *p* < 0.05) identified in the NCT/RBC2 line contrast at 43 °C, the greatest predicted fold change was observed for the GO Biological Process categories *Negative regulation of cellular amide metabolic process* (GO:0034249), *Negative regulation of translation* (GO:0017148)*,* and *translation* (GO:0006412) (Table 4). Up-regulated DEGs in the NCT SCs relative to the RBC2 cells showed overrepresentation in *Post-transcriptional gene silencing* (5.78-fold, *p* = 4.00 × 10^−02^), *Peptidyl-serine phosphorylation* (3.11x, *p* = 8.08 × 10^−04^), and *Peptidyl-serine modification* (2.91x, *p* = 2.89 × 10^−03^), whereas down regulated genes showed greatest overrepresentation in *Protein refolding* (6.38x, *p* = 3.05 × 10^−02^), *Cytoplasmic translation* (5.16x, *p* = 2.09 × 10^−05^) and *Translation* (4.78x, *p* = 3.77 × 10^−35^). Activation z-scores in IPA were generally positive and included several signaling pathways. The highest positive activation score was observed for *Cardiac hypertrophy signaling* (z = 4.38, −log(pval) = 8.24) and lowest z-score was observed for the *Oxidative phosphorylation pathway* (z = −6.0, −log(pval) = 5.97) (Figure 8c). Differential expression of genes in the pathways with greatest |z|-score included a greater percentage of up-regulated genes.

#### 3.3.2. Line Differences: NCT Versus F-Line SCs

Although different genome annotations were used for primary analyses, a comparison was made of the responses of the differentiating 1-wk SCs in the present study (NCT) to those of the 7-wk SCs previously examined (F-line) [25]. A composite gene list was created using gene IDs common to the two gene lists (*n* = 11,565). For the cold-treated cell comparison, the combined gene list included 279 DEGs (FDR *p*-value < 0.05 and |Log_2_FC| > 2.0) in the NCT SCs and 420 DEGs in the F-line cells (Appendix A). Comparison of the lists of DEGs, found 128 unique to NCT and 269 unique to the F-line; 151 DEGs were shared. The majority of unique genes were down regulated in the NCT cells (54%) as was the case for the DEGs of the F-line SCs (70%). The 151 shared DEGs were primarily down regulated (92%) and directionality and magnitude of the fold changes were similar between lines. GO analysis of the DEGs from the cold-treated cell comparison indicates association of these genes with myofibril assembly, and muscle development and contraction.

For the heat-treated cell comparison, the combined gene list included 193 DEGs (*p*-value < 0.05 and |Log_2_FC| > 2.0) in the NCT SCs and 175 DEGs in the F-line cells (Appendix A). Comparison of the lists of DEGs found 160 unique to NCT and 142 unique to the F-line; 33 DEGs were shared. The majority of unique genes were up regulated in the NCT cells (61%) as was the case for the F-line SCs (67%). GO analysis of the unique DEGs indicates association with cell adhesion. The 33 shared DEGs had slightly more down-regulated genes (58%) and like the cold-treated cell comparison, the directionality and magnitude of the fold changes were similar between lines. These genes also show enrichment associated with regulation of cell-substrate adhesion.

Next, the 11,565 genes common in the two studies were mapped in IPA (*n* = 9961) and were used to run a Comparison analysis (33 °C versus 38 °C and 43 °C versus 38 °C) using the FDR *p*-values and Log_2_FC as variables. Comparison analysis of DEGs of the NCT SCs to those of the F-line (16-wk bodyweight selected) found the significant differences in activation (z-score) both between lines (within temperature) comparisons and between temperature comparisons for several canonical pathways (Figure 9). These comparisons identify genes responding to thermal stress that are different between the 1-wk NCT SCs and the 7-wk F-line cells. Expression differences in these genes may be attributed to both genetic differences between the bird lines and the age of the birds from which the SCs were isolated (1 wk versus 7 wk). Studies of the developmental cascade in turkey skeletal muscle suggest that differentiation potential of skeletal muscle SCs significantly decreases by 4 wk of age [5].

## 4. Discussion

Embryonic muscle growth occurs through the myoblast leading to muscle fiber formation by hyperplasia. Posthatch hypertrophic muscle growth involves the proliferation, differentiation, and fusion of SCs with existing myofibers. Satellite cells are a heterogeneous population of cells with varying proliferation and differentiation capacities [51] and their activity during this time is controlled by growth factors, intracellular signaling pathways and transcription factors. Key in modulating the cellular microenvironment is antagonistic Notch and Wnt signaling. The membrane receptor Notch, activated by the ligand Delta (DLL1), is involved in initiating proliferation. Canonical Wnt/β-catenin pathway regulates the SCs transition from cell proliferation to differentiation through the membrane receptor Frizzled [11,52]. As the downstream target of MyoD, DLL1 has been reported to promote myogenic differentiation via the MyoD/DLL1/Notch gene axis [53]. Comparison of RNA-seq data between the differentiating cells and proliferating cells of our previous study found expression of myogenic marker genes including *MyoD* and *MyoG* was higher in the turkey SCs after 48 h of differentiation consistent with their progressive development [26]. 

The nuclear receptor peroxisome proliferator-activated receptor-γ (PPARγ) is required for induction of adipogenic differentiation of myogenic cells [54]. As reported by Xu et al. [55], differentiation in both RBC2 and NCT satellite cells is significantly affected by thermal stress. In the differentiating turkey SCs, *PPARγ* was significantly down regulated, consistent with previous results suggesting that adipogenic potential of the turkey SCs is higher during proliferation and gradually decreases with differentiation [55]. Also down regulated was *MSTN* (myostatin) and *MYF6* (*MRF4* or herculin). Early muscle growth requires a balance between the proliferation of new precursor cells and differentiation into muscle fibers. This process is tightly regulated by growth factors such as myostatin, a key negative regulator of muscle growth that, in turn, down regulates the regulatory factors MyoD and MyoG, thereby balancing cell proliferation and differentiation [56]. In birds, MSTN has different isoforms resulting from the alternative splicing of *MSTN* mRNA. MSTN-A is anti-myogenic isoform whereas MSTN-B is pro-myogenic [57]. The relative abundance of these two protein isoforms in the differentiating turkey SCs is unknown, but may be pro-myogenic. Down regulation of *MYF6* may seem contraindicated however, its expression is typically associated with older myotubes where it is thought to promote myofiber maturation and maintenance of the differentiated state [58]. Recent studies in mice also indicate a novel role for Myf-6 in promoting myokine expression to block premature differentiation, preventing stem cell exhaustion [59]. Further study of these proteins in birds is warranted.

The only gene that was consistently up regulated at all temperatures in the NCT SCs compared to the RBC2 cells was *ETNPPL* (ethanolamine-phosphate phospho-lyase). This gene functions in phospholipid metabolism and is associated with fatty acid and lipid synthesis [60,61]. Although a significant decrease in lipid content was observed in both NCT and RBC2 cells at 24 h of differentiation under cold stress (33 °C) [55], the differential response in *ETNPPL* between the lines supports observations that growth selection has increased the adipogenic potential of SCs, increasing intramuscular fat deposition in the faster-growing turkeys.

Of the 15 genes consistently down regulated at all the temperatures in the NCT SCs compared to RBC2 cells, three of them (*ENSMGAG00000022282*, *KCMF1*, and *LOC100303669*) are involved in the ubiquitin proteasome pathway. This suggests that breast muscle in NCT birds has a reduced potential for degeneration and atrophy through the ubiquitin proteasome pathway compared to the RBC2 line. Also consistently down regulated was *NIPBL* (*LOC100544159*, nipped-B-like protein), a gene associate with cell cycle progression [62]. Down regulation of *NIPBL* may induce cell cycle arrest and would thus potentially act to delay myogenesis in the NCT SCs during differentiation.

### 4.1. Effect of Cold Stress

Cold treatment resulted in significant gene expression changes in the SCs from both turkey lines, with the primary effect being down regulation of affected genes (almost 3-fold). A larger number of DEGs were seen in the NCT cells, consistent with our previous study where a greater number of genes were affected by cold treatment in the 7-wk SCs from the growth-selected F-line than the Randombred RBC2 line [25]. Cold stress in differentiation has also been shown to down regulate the mTOR and Wnt/PCP pathways to a greater extent in NCT SCs compared to the RBC2 cells [63,64]. 

Clearly the NCT and RBC2 SCs have different responses during cold stress. Notably, NF-κB, a key stimulator of muscle atrophy and inflammatory myopathies [65], is differentially affected by cold only in the NCT cells. Changes in NF-κB signaling may affect muscle atrophy and cause inflammatory myopathies. Under cold stress, DEGs involved in regulation of skeletal muscle tissue regeneration and sarcomere organization were significantly overrepresented with overall down regulation of muscle-associated genes (*MYH*) in both cell lines. Interruption of muscle fiber assembly and sarcomere organization by the cold stress would slow differentiation resulting in smaller myotubes (decreased width) as seen in previous studies [66]. Still, expression of genes involved in muscle sarcomere organization, myofibril assembly, and striated muscle cell development remained significantly higher in the NCT SCs compared to the RBC2 cells at the cold (33 °C) and control (38 °C) temperatures. These results reflect the effect of growth selection on increasing myogenic potential (myotube formation) and myofiber hypertrophy. 

Altered expression of the oxidative phosphorylation pathway was predicted in the SCs under cold stress and show higher predicted activation in the NCT SCs compared to the RBC2 cells at control and cold temperatures. Altered oxidative phosphorylation may result in oxidative stress, which can lead to oxidative muscle damage [67,68]. This suggests that NCT SCs may have increased oxidative potential relative to the RBC2 cells, important in limiting the incidence of myopathies associated with oxidative stress. Predicted effects on oxidative stress pathways have also been observed in cold-stressed turkey poults [69].

### 4.2. Effect of Heat Stress

A larger suite of genes were significantly affected by heat treatment (43 °C) than cold (33 °C) reflecting the sensitivity of the SCs to heat stress. Many cell surface protein receptors are down regulated including *NTSR1*, associated with fat production in skeletal muscle [70], *ADGRL4*, that stimulates cell cycle progression, and *ADRA1D*, that regulates myoblast survival and differentiation [71]. Down regulation of these genes may decrease the differentiation, survival and fat production of cells particularly in the RBC2 SCs. 

Heat stress resulted in the up regulation of several genes that encode contractile proteins (*MYH1E*, *MYBPC1*, *MYH2*, myosin motor domain-containing protein), particularly in the RBC2 cells. Commercial birds grow larger muscle fibers than the RBC2 line, and therefore increase in the contractile protein myosin is expected. However, an increase during heat stress suggests promotion of muscle growth. Also included in the affected genes were DEGs associated with sarcomere assembly. For example, in the NCT cells, CLASP2, involved in microtubule organization and cell adhesion [72], was significantly up regulated suggesting enhanced myotube formation (migration and fusion) during heat stress. Greatly up regulated in the NCT cells were genes reported to regulate myoblast survival and differentiation (ADRA1D, [71]) and promote cell adhesion (FREM1, [48]). Their increased expression in SCs may promote increased hypertrophic potential of the *p. major* muscle and sarcomere assembly in the commercial turkeys. 

The proliferation and differentiation of turkey SCs under thermal stress also provides insight into the regeneration ability of muscle and the myopathies that negatively affect poultry. For example, Wooden Breast is a necrotic/fibrotic myopathy that negatively impacts breast meat quality in chickens with meat product downgrades or condemnation. In commercial broilers affected with Wooden Breast, sarcomere organization is reduced, especially in small and intermediate-sized fibers [73]. Irregular, disconnected, and reduced sarcomeric Z bands are observed by 6 wk of age in Wooden Breast-affected muscles [74]. These data suggest that the SC-mediated repair process of necrotic fiber damage in broilers is not able to restore sarcomeric muscle fiber structure. However, there are no reports of necrotic/fibrotic myopathies like Wooden Breast in modern commercial meat-type turkeys, although these heavy-weight birds are in part, selected for *p. major* muscle growth. 

Although growth selection strategies in turkeys and broilers have shared similarities, the outcome on breast muscle growth, development, and structure are distinct. The difference between current meat-type broilers and turkeys appears to at least partially involve differences stemming from the SCs of the *p. major* muscle that are responsible for the post-hatch hypertrophic growth and muscle regeneration. Selection for breast growth in chickens has created birds with greater muscle hypertrophy that respond differently to environmental stressors and may exhaust SC pools leading to diminished SC function [75]. Prior to the onset of Wooden Breast, current broilers may undergo decreased SC proliferation and differentiation (Velleman, unpublished observation). In contrast, Xu et al. [66] has shown an increase in proliferation and differentiation of current commercial turkey SCs compared to cells from a line of birds established prior to and without intense growth selection. Data from the current study shows that expression of genes promoting muscle regeneration (differentiation) and sarcomere assembly is higher in SCs from faster-growing modern commercial turkeys compared to slower-growing historic turkeys. Taken together, these findings may explain why necrotic/fibrotic myopathy like Wooden Breast is not found in modern-commercial turkeys.

## 5. Conclusions

Timing of thermal stress during poultry production can differentially affect bird performance and production traits. Movement of poults from hatch to grower facilities during the period where the thermal regulatory system is poorly developed, can expose them to acute thermal conditions, either hot or cold, that may have long-term consequences for muscle development and ultimately meat quality. Understanding how temperature affects SC proliferation and differentiation will allow for development of better thermal management strategies especially to confront the uncertainties brought about by climate change. Growth selection appears to have increased the proliferation, differentiation, adipogenic potential, oxidative potential, and decreased the degenerative and atrophic potential of the turkey *p. major* SCs. The breast muscle of fast-growing meat-type birds is characterized by its excessive hypertrophic myofiber growth potential with decreased incidence of muscle atrophy and degeneration. The biology of the SCs and their effect on morphological structure of the breast muscle is of continued importance in the poultry industry, especially in light of emerging myopathies [76].

## Figures and Tables

**Figure 1 genes-13-01857-f001:**
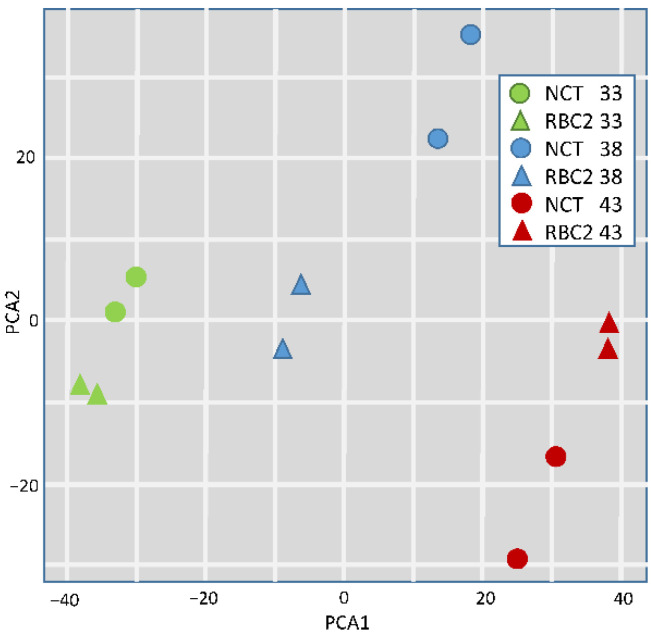
Principal component analysis (PCA) plot of normalized RNAseq read counts. Sample to sample distances (within- and between-treatments) are illustrated for each dataset on the first two principal components. Samples are plotted according to treatment.

**Figure 2 genes-13-01857-f002:**
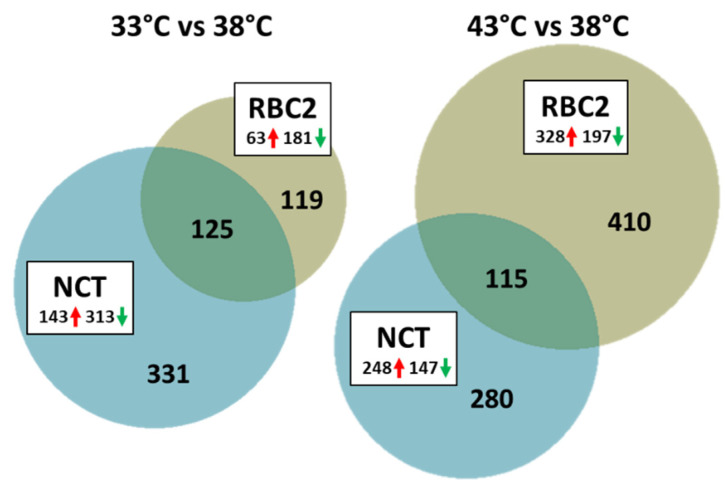
Distribution of differentially expressed genes during differentiation of cultured turkey *p. major* SCs. For each temperature comparison, the number of genes with FDR *p*-value < 0.05 and |Log_2_FC| > 2.0 that were shared or unique to each line (RBC2 and NCT) are indicated in the Venn diagram. Circle size is proportional to the number of DEGs. Directional expression of genes is indicated by red (up) or green (down).

**Figure 3 genes-13-01857-f003:**
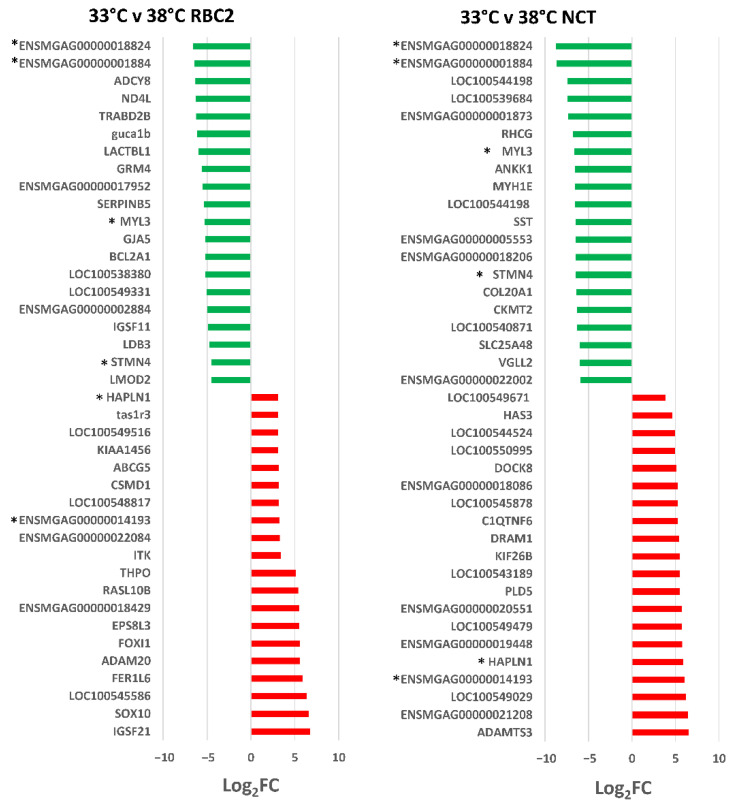
Directional change in the genes showing the highest expression differences in comparison between lines (RBC2 and NCT) at 33 °C and 38 °C. For each with-in temperature comparison the Log_2_FC is plotted is plotted for each DEG (FDR *p*-value < 0.05). Genes shared between temperatures are indicated by asterisks (*). Directional expression of genes is indicated by red (up) or green (down).

**Figure 4 genes-13-01857-f004:**
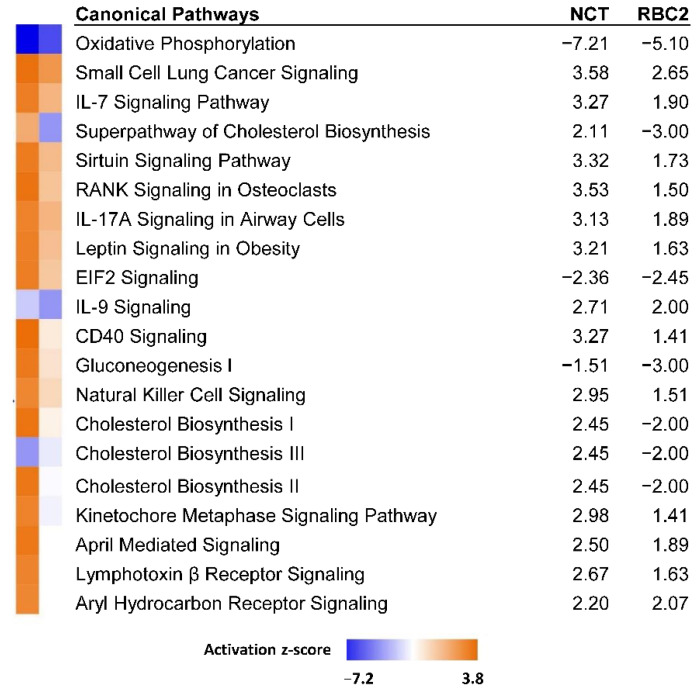
Significant canonical pathways identified in IPA Comparison analysis of genes expressed in NCT versus RBC2 turkey skeletal muscle SCs at 33 °C. Only the 20 pathways with highest composite |z|-scores are shown.

**Figure 5 genes-13-01857-f005:**
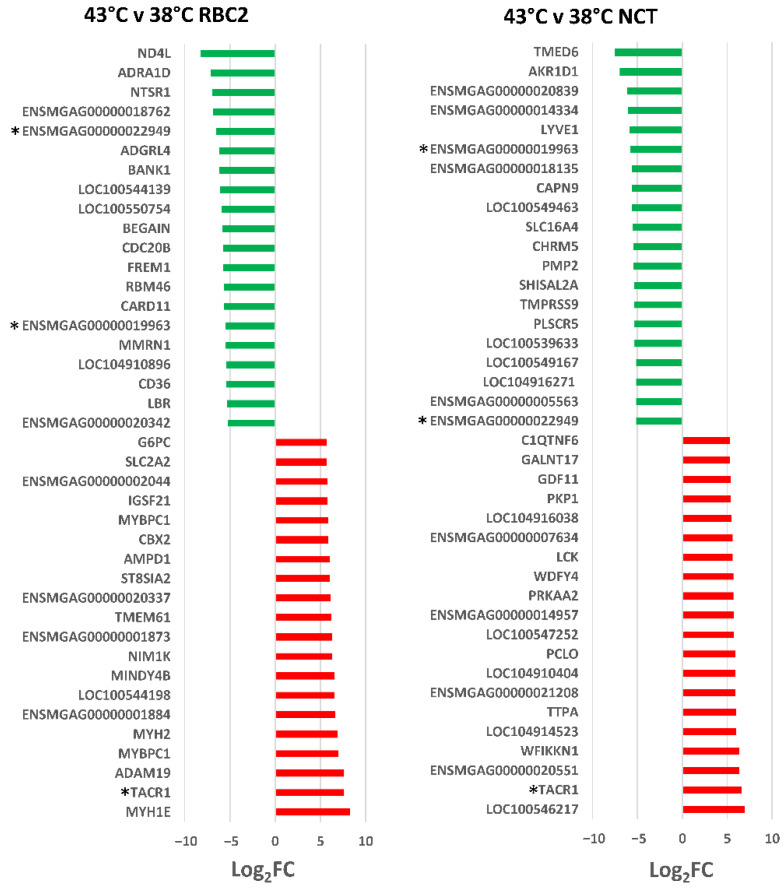
Directional change in the genes showing the highest expression differences in comparison between lines (RBC2 and NCT) at 43 °C and 38 °C. For each with-in temperature comparison the Log_2_FC is plotted is plotted for each DEG (FDR *p* value < 0.05). Genes shared between temperatures are indicated by asterisks (*). Directional expression of genes is indicated by red (up) or green (down).

**Figure 6 genes-13-01857-f006:**
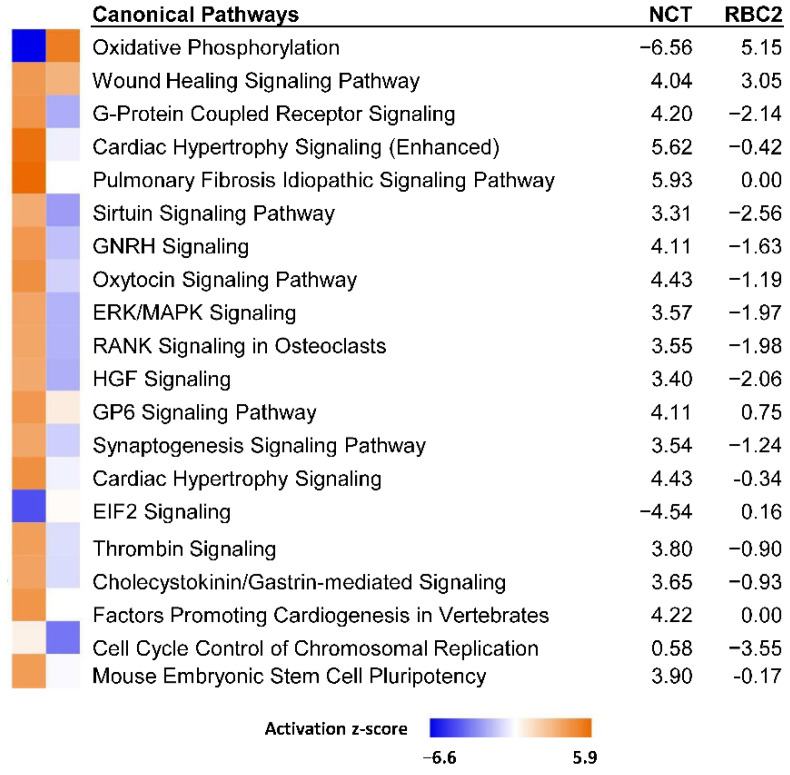
Significant canonical pathways identified in IPA Comparison analysis of genes expressed in NCT versus RBC2 turkey skeletal muscle SCs at 43 °C. Only the 20 pathways with highest composite |z|-scores are shown.

**Figure 7 genes-13-01857-f007:**
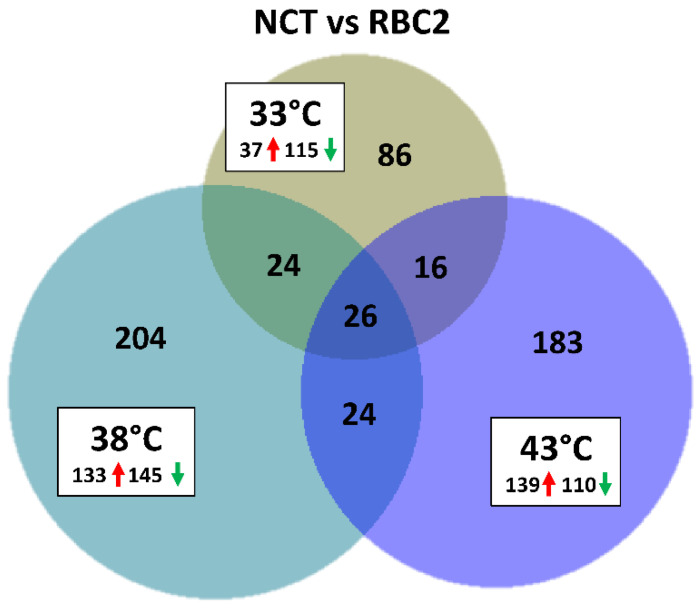
Distribution of differentially expressed genes between lines (NCT vs. RBC2) during *p. major* SC differentiation. For each temperature comparison, the number of genes with FDR *p*-value < 0.05 and |Log_2_FC| > 2.0 that were shared or unique to each incubation temperature are indicated. The number and direction of expression change (red,↑ up or green, ↓ down) for the genes included in each temperature group are listed outside the Venn diagram. Circle size is proportional to the number of DEGs.

**Figure 8 genes-13-01857-f008:**
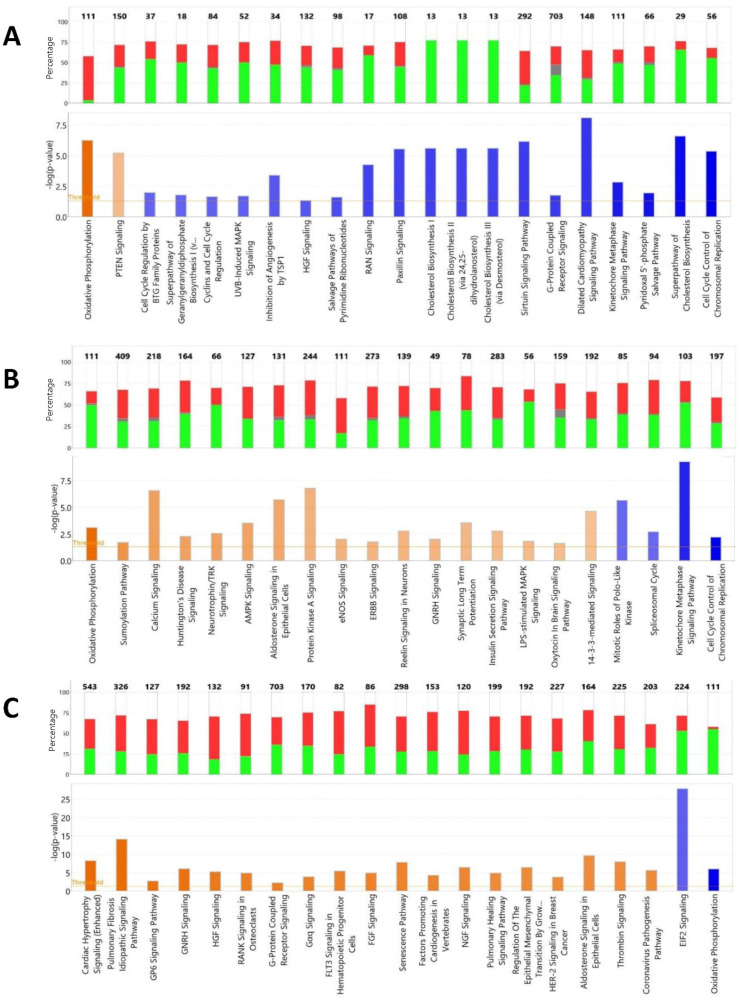
Summary of IPA Pathway Activity Analysis of DEGs observed comparison of NCT and RBC2 SCs differentiated at 38 °C (**A**), 33 °C (**B**) or 43 °C (**C**). Pictured are the number of affected molecules, gene expression pattern and z-score for the significant canonical pathways with the greatest absolute z-score. Directional expression of genes is indicated by red (up) or green (down) and relative z-score by color intensity (orange = positive z and blue = negative z).

**Figure 9 genes-13-01857-f009:**
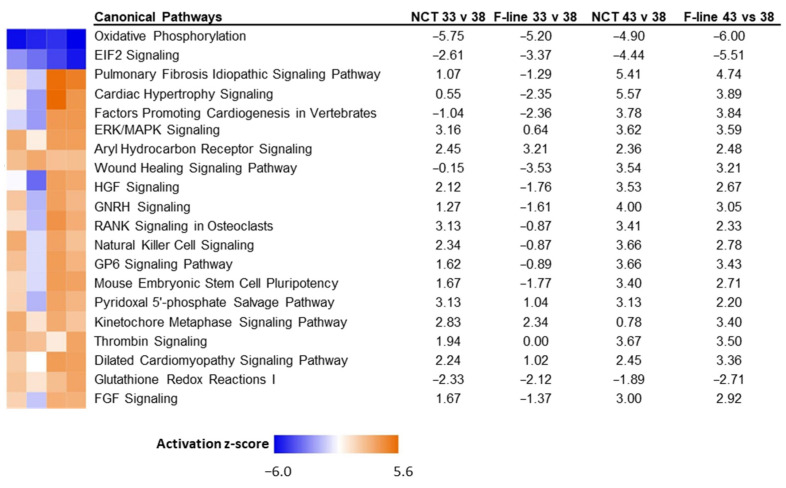
Significant canonical pathways identified in IPA Comparison analysis of genes expressed in skeletal muscle SCs of NCT versus F-Line at 33 °C and 43 °C compared to controls (38 °C). Only the 20 pathways with highest absolute z-scores are shown.

**Table 1 genes-13-01857-t001:** Summary of differentiation RNAseq data.

Line	Temp °C	Replicate	PE Reads	Median Read Quality R1	Median Read Quality R2	Observed Genes	Mean Observed Genes	Group Observed Genes	% Expressed Genes
RBC2	33	1	22,142,413	36.2	35.8	13,926	13,917.5	14,459	0.805
		2	24,577,416	36.3	35.8	13,909			
	38	1	19,480,565	36.2	35.8	13,638	13,732.5	14,295	0.795
		2	21,879,398	36.2	35.8	13,827			
	43	1	20,058,265	36.3	35.8	13,589	13,612.5	14,184	0.789
		2	21,823,851	36.2	35.8	13,636			
NCT	33	1	25,893,254	36.2	35.9	13,816	13,763.0	14,309	0.796
		2	21,843,150	36.2	35.8	13,710			
	38	1	19,234,626	36.2	35.8	13,537	13,569.0	14,143	0.787
		2	24,062,238	36.3	35.8	13,601			
	43	1	27,102,323	36.2	35.9	14,000	13,917.5	14,495	0.807
		2	25,216,803	36.3	35.8	13,835			
Mean			22,776,191.8	36.23	35.82	13,752.0	13,752.00	14,314.2	0.7966

For each library the total number of raw paired-end reads, median read qualities (R1 and R2), the number of observed genes (mapped reads > 1) by library and treatment group and the percentage of expressed genes (group read count > 1.0) are given.

**Table 2 genes-13-01857-t002:** Summary of gene expression and significant differential expression (DE) in pair-wise comparisons of differentiating cells.

Comparison	Groups	Total Expressed Genes	Shared Genes	Unique Genes by Group	FDR < 0.05	|Log_2_FC| > 1.0	|Log_2_FC| > 2.0
Cold (33 °C)	33R vs. 38R	14,132	13,224	527/381	3621	814	244
	33N vs. 38N	14,013	13,045	540/428	5600	1914	456
Hot (43 °C)	43R vs. 38R	14,072	13,075	467/530	6248	2043	525
	43N vs. 38N	14,124	13,106	651/367	4137	1795	395
Line							
	38N vs. 38R	14,027	13,051	422/554	3777	1357	278
	33N vs. 33R	14,129	13,207	378/544	2919	501	152
	43N vs. 43R	14,150	13,149	608/393	4574	1365	249

For each comparison of the treatment groups (Temperature: 33 °C, 38 °C or 43 °C, Line: RBC2 = R or NCT = N), the total number of expressed and uniquely expressed genes (genes expressed uniquely in each group comparison), the number of genes with significant False Discovery Rate (FDR) *p*-value, and the numbers of significant genes also with |Log_2_FC (fold change)| > 1.0 and > 2.0 are given. Only those genes with treatment group mean normalized read counts > 1.0 are counted as expressed.

**Table 3 genes-13-01857-t003:** Differential expression of skeletal muscle marker genes in SCs following 48 h of differentiation at 38 °C compared to 72 h of proliferation [26]. For each gene, the directional change in gene expression between differentiating and proliferating cells, the fold change (Log_2_FC) and False Discovery Rate (FDR) *p*-value are given for the comparisons within each turkey line (RBC2 and NCT). Directional expression of genes is indicated by red (up) or green (down).

Gene ID	Line	Change	Log_2_FC	FDR *p*-Value Correction
MYOD1, myogenic differentiation 1	RBC2	↑	3.521	8.59 × 10^−157^
	NCT	↑	1.873	2.72 × 10^−13^
PAX7, Paired box 7	RBC2	↑	0.808	5.22 × 10^−08^
	NCT	↑	0.946	6.71 × 10^−03^
MYOG, myogenin	RBC2	↑	8.387	0.00
	NCT	↑	6.695	1.77 × 10^−113^
DLL1 delta like canonical Notch ligand 1	RBC2	↑	3.835	1.31 × 10^−46^
	NCT	↑	3.524	1.41 × 10^−18^
MSTN, myostatin	RBC2	↑	1.595	1.77 × 10^−04^
	NCT	↑	2.519	6.23 × 10^−08^
MYF6, myogenic factor 6	RBC2	↓	−3.104	0.00
	NCT	↓	−2.502	3.09 × 10^−18^
PPARγ, peroxisome proliferator activated receptor γ	RBC2	↓	−2.333	2.03 × 10^−108^
	NCT	↓	−2.228	1.96 × 10^−13^
LIF, LIF interleukin 6 family cytokine	RBC2	↓	−3.416	4.24 × 10^−58^
	NCT	↓	−2.651	1.88 × 10^−08^

**Table 4 genes-13-01857-t004:** PANTHER Overrepresentation test of DEGs in the NCT cells relative to the RBC2 after 48 h of differentiation. Shown are the gene ontology categories with the greatest fold enrichment in the GO biological process category ^1^.

	*Gallus gallus* (18109)	Turkey DEGs	Expected	Fold Enrichment	*p*-Value
**Control 38 °C (3777 DEGs, 2915 uniquely mapped)**					
striated muscle cell development (GO:0055002)	45	27	7.24	3.73	2.37 × 10^−03^
myofibril assembly (GO:0030239)	44	26	7.08	3.67	4.76 × 10^−03^
negative regulation of protein depolymerization (GO:1901880)	48	25	7.73	3.24	4.15 × 10^−02^
cellular component assembly involved in morphogenesis (GO:0010927)	58	29	9.34	3.11	1.52 × 10^−02^
regulation of protein depolymerization (GO:1901879)	60	29	9.66	3.00	3.33 × 10^−02^
regulation of protein-containing complex disassembly (GO:0043244)	77	37	12.39	2.99	1.70 × 10^−03^
actomyosin structure organization (GO:0031032)	79	36	12.72	2.83	8.95 × 10^−03^
muscle cell development (GO:0055001)	84	38	13.52	2.81	4.57 × 10^−03^
negative regulation of supramolecular fiber organization (GO:1902904)	77	34	12.39	2.74	2.50 × 10^−02^
striated muscle cell differentiation (GO:0051146)	102	44	16.42	2.68	1.80 × 10^−03^
**Cold 33 °C (2919 DEGs, 2343 uniquely mapped)**					
striated muscle cell development (GO:0055002)	45	21	5.82	3.61	4.06 × 10^−02^
stem cell development (GO:0048864)	57	24	7.37	3.25	3.79 × 10^−02^
striated muscle cell differentiation (GO:0051146)	102	42	13.20	3.18	3.00 × 10^−05^
muscle cell development (GO:0055001)	84	34	10.87	3.13	1.12 × 10^−03^
regulation of axonogenesis (GO:0050770)	68	27	8.80	3.07	2.52 × 10^−02^
ameboidal-type cell migration (GO:0001667)	91	36	11.77	3.06	7.85 × 10^−04^
muscle cell differentiation (GO:0042692)	129	50	16.69	3.00	7.37 × 10^−06^
mesenchymal cell differentiation (GO:0048762)	103	38	13.33	2.85	1.67 × 10^−03^
stem cell differentiation (GO:0048863)	87	32	11.26	2.84	1.56 × 10^−02^
muscle structure development (GO:0061061)	201	73	26.01	2.81	1.28 × 10^−08^
**Hot 43 °C (4573 DEGs, 3885 uniquely mapped)**					
negative regulation of cellular amide metabolic process (GO:0034249)	99	47	19.45	2.42	1.27 × 10^−02^
negative regulation of translation (GO:0017148)	95	45	18.67	2.41	2.43 × 10^−02^
translation (GO:0006412)	290	133	56.98	2.33	6.42 × 10^−11^
peptide biosynthetic process (GO:0043043)	298	133	58.55	2.27	3.35 × 10^−10^
peptide metabolic process (GO:0006518)	377	156	74.07	2.11	3.04 × 10^−10^
amide biosynthetic process (GO:0043604)	374	147	73.48	2.00	3.76 × 10^−08^
regulation of translation (GO:0006417)	198	76	38.90	1.95	1.08 × 10^−02^
positive regulation of catabolic process (GO:0009896)	197	74	38.71	1.91	2.34 × 10^−02^
regulation of cellular amide metabolic process (GO:0034248)	213	80	41.85	1.91	1.08 × 10^−02^

^1^ Turkey DEGs were matched to the chicken reference gene list IDs for analysis in PANTHER. For each temperature, the number of genes in the reference list and turkey DEGs are given. *p*-values are as determined by Fisher Exact Test with Bonferroni correction.

## Data Availability

The datasets presented in this study are accessioned in the NCBI’s Gene Expression Omnibus (GEO) repository as part of SRA BioProject PRJNA842679; https://www.ncbi.nlm.nih.gov/genbank/, PRJNA842679 (accessed 8 September 2022).

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
