# Peer review of "Transcriptome Response of Differentiating Muscle Satellite Cells to Thermal Challenge in Commercial Turkey"

_genes, 2022, doi:10.3390/genes13101857_

Round 1

Reviewer 1 Report

Manuscript Genes-1924035 entitled “Transcriptome Response of Differentiating Muscle Satellite
Cells to Thermal Challenge in Commercial Turkey. Please notice the following:

General view: The manuscript discussed a novel and great idea in excellent language and grammar. The manuscript was appropriately expressed.

The manuscript could be accepted for publication after minor revision.

Title: Clear, informative, and concise.

Abstract: Clear to a high degree but notice that at L12-17: The problem background is some sort long and needs to be concise.

Introduction: Improperly expressed. The introduction section has to be rearranged into three paragraphs only i.e., 1. Introduction 2. Significance of the study, and 3. Aim of the study.

The aim: Clear and informative.

Materials and Methods: Please notice the following:

1.      The authors did not list the institutional approval for the study.

2.      Please provide the statistical model used in the study ANOVA.

Results: Novel, clear, and informative. Please consider listing the p-value in this section after the “significant and highly significant” few expressions in the results section.

Discussion: Clear, informative, properly expressed, and contribute to knowledge with a good level of speculations and a moderate level of comparison.

Conclusion: Clear and informative.

Authors’ contributions: Clear and informative.

Funding: Clear and informative.

Acknowledgment: Clear and informative.

References: MUST BE UPDATED as only 27.6% (21 out of 76) were published in the past five years. This percentage has to be increased to at least 30-40%.

Tables: Well organized and presented.

Figures: Well organized and presented.

Author Response

We thank the reviewer for the positive comments on our manuscript.  The reservations of this review are addressed below.

Abstract: Clear to a high degree but notice that at L12-17: The problem background is some sort long and needs to be concise.

The implication of this comment is unclear to us. We believe these few introductory sentences are necessary to describe the scope and significance of the work outlined in the abstract.

Introduction: Improperly expressed. The introduction section has to be rearranged into three paragraphs only i.e., 1. Introduction 2. Significance of the study, and 3. Aim of the study.

The Instructions to Authors states the following: The introduction should briefly place the study in a broad context and highlight why it is important. It should define the purpose of the work and its significance, including specific hypotheses being tested. The current state of the research field should be reviewed carefully and key publications cited. Please highlight controversial and diverging hypotheses when necessary. Finally, briefly mention the main aim of the work and highlight the main conclusions. Keep the introduction comprehensible to scientists working outside the topic of the paper.  There is no indication that the Introduction must be limited to 3 paragraphs. Indeed, we feel that we have balanced brevity with appropriate depth necessary for the reader to address the journal’s instructions.

Materials and Methods: Please notice the following:

  1. The authors did not list the institutional approval for the study.

This study utilized cells previously isolated and used for prior investigations.  No institutional approval is necessary of these cell culture studies.

  1. Please provide the statistical model used in the study ANOVA.

 This has been updated in the text.

Results: Novel, clear, and informative. Please consider listing the p-value in this section after the “significant and highly significant” few expressions in the results section.

P values have been added throughout the text as appropriate.

References: MUST BE UPDATED as only 27.6% (21 out of 76) were published in the past five years. This percentage has to be increased to at least 30-40%.

This is a somewhat unexpected comment.  We have made every attempt to cite the literature relevant to our study including those of both historical significance and several papers published in the past 2 years.  There is no requirement stated in the Instructions to Authors nor precedence that a percentage of the combined references published in the past 5 years meet a certain value.

Reviewer 2 Report

The manuscript describes global transcriptional changes in breast muscle satellite cells isolated from 1-week-old turkeys cultured in cold (33C), normal (38C), or hot (43C) temperatures. This is an important topic, as during the early post-hatch period the birds’ thermoregulatory systems are not fully developed and they are susceptible to cold or heat stress. As this period is also when muscle satellite cells are rapidly dividing, any perturbations resulting from cold or hot thermal challenges could have lasting effects on muscle growth and development. The paper is well written, with a large amount of data presented in a clear and concise manner. The lack of biological replication is a major limitation of the study, and clarification as to how temperature treatments were chosen should be provided. Specific revisions related to improving descriptions of methodology, data inclusion and presentation, and relevance of the discussion to the findings in the paper should also be made.   

- Virtually no detail is provided for the methodology used in the study (e.g. satellite cell isolation & culture, RNAseq, bioinformatics), though references to the authors’ previous work are provided. It would greatly improve readability if at least brief descriptions of methods used were provided so readers can use this information to facilitate their own interpretations of results without having to look up multiple papers.

-  Line 85: The referenced paper appears to relate to satellite cell isolation from 7-week-old birds, not the 1-week-old birds that were used in the present study. Clarify whether cells used in this study were just isolated using similar methodology, or correct the reference provided.

- Lines 92 – 99: Thermal challenges were applied during differentiation with reduced-serum media. Data (including images, if possible) related to how differentiation was influenced by thermal challenge should be provided.

- Lines 96 – 97: Provide some justification for the temperatures chosen as “cold stress,” “control,” and “heat stress.” Were they chosen in relation to the bird’s body temperature (in which case an argument can be made for the “control” treatment being cold and “heat stress” treatment being considered closer to normal) or to represent ambient temperature or brooding conditions that would be considered cold, optimal, and hot (in which case, the actual muscle tissue might not be exposed to those temperatures).

- Line 118: As described, the study does not appear to have any biological replication. Only technical replication by plating duplicate wells from the same pool of cells was conducted. If that is the case and there is some reason the study was not or cannot be biologically replicated, this does need to be explained and acknowledged as a limitation. Ways in which individual variation was captured in the data set should be provided (e.g. Were the satellite cells at least a pool from multiple individuals? Were they from one sex or mixed sex? Why was the experiment not able to be replicated in time or with multiple pools of cells?) 

- Some table titles (in paper and supplemental) and figure legends refer to “F-line” cells and not NCT cells when it is clear they are referring to cells used in this experiment, which I think are different from “F-line cells”. This needs to be corrected throughout. If I am misunderstanding the “F-line” is synonymous the “NCT”, the line should be consistently referred to.

- Most figures and tables (including supplemental) need additional information so they can stand-alone. For example, in Table 2 “R” and “N” are used as abbreviations for the lines and not defined and it is not immediately clearly by the layout of the table how the “Unique Genes by Group” column is set up. There are also frequent abbreviations used in tables or figures that are not defined (e.g. FDR, FC, Diff, Pro, etc) and color coding in supplemental tables that is not described. While many of these things can be ultimately inferred, it is a better practice to have tables & figures fully able to stand-alone. All figures/tables need to be reviewed and changed as necessary.

-Supplemental Table 1:  No standard deviation is given as is stated in the title, so this needs to be added. It is not clear why the “overall mean” column is included, but this would be better titled as “mapped reads/sample”.

- Supplemental Table: The title does not describe information presented in the table and needs to be corrected. It might also be helpful to provide additional separate tabs for each comparison that only include differentially expressed genes that are sorted by fold difference.

- Lines 462 – 466 and lines 494 – 499 have identical text yet are describing different data. Assume this was a copy/paste oversight, but this needs to be corrected. What the numbers after each categories mean and where they were derived from needs to be provided, since they do not appear to come from Table 4.

- Lines 542 – 546, Supplementary Tables 3 & 4: It is unclear why this comparison was made, so a rationale for including it needs to be provided. It is acknowledged that analysis for each were done using different genome annotations, and both line and age are different so confound any ability to attribute differences in response between the groups to thermal challenge. The authors need to provide a stronger justification or remove this section.

- Lines 683 – 703: This section of the discussion seems entirely irrelevant to the data presented or the premise of the study as written. It is unclear why it is included, so either should be removed or a stronger connection made with data presented.

Author Response

We thank the reviewer for the thorough review of our manuscript and point to several items to address.  We detail our response to each of these in bold text below.  Addressing the reservations identified have greatly improved our manuscript.

Concerns:

- Virtually no detail is provided for the methodology used in the study (e.g. satellite cell isolation & culture, RNAseq, bioinformatics), though references to the authors’ previous work are provided. It would greatly improve readability if at least brief descriptions of methods used were provided so readers can use this information to facilitate their own interpretations of results without having to look up multiple papers.

We have expanded the methods section to briefly summarize the methodology used and cited in the text. 

-  Line 85: The referenced paper appears to relate to satellite cell isolation from 7-week-old birds, not the 1-week-old birds that were used in the present study. Clarify whether cells used in this study were just isolated using similar methodology, or correct the reference provided.

The paper referenced is the protocol used to isolate the cells. The cells used in this study were from 1-wk old birds isolated using similar methodology.  These are the same cell lines used in other experiments by our group and cited in the manuscript (Xu et al., 2021).

- Lines 92 – 99: Thermal challenges were applied during differentiation with reduced-serum media. Data (including images, if possible) related to how differentiation was influenced by thermal challenge should be provided.

As reported by Xu et al. (2021), thermal stress of the RBC2 and NCT satellite cells significantly affects their differentiation.  This statement has been added to the discussion.   

     Reference:

     Xu, J., Strasburg, G.M., Reed, K.M., Velleman, S.G. Response of turkey pectoralis major muscle satellite cells to hot 881 and cold thermal stress: Effect of growth selection on satellite cell proliferation and differentiation. Comp Biochem 882 Physiol A Mol Integr Physiol. 2021, 252, 110823. 883

- Lines 96 – 97: Provide some justification for the temperatures chosen as “cold stress,” “control,” and “heat stress.” Were they chosen in relation to the bird’s body temperature (in which case an argument can be made for the “control” treatment being cold and “heat stress” treatment being considered closer to normal) or to represent ambient temperature or brooding conditions that would be considered cold, optimal, and hot (in which case, the actual muscle tissue might not be exposed to those temperatures).

We realize that the actual muscle tissue might not be exposed to the “cold” and “hot” range used in this study, and this is always one of the limitations of in vitro studies. The experimental temperatures were as determined and used in our prior studies. The control temperature of 38°C is approximately that measured in newly hatched turkey poults (38.0 - 38.5°C, Strasburg, unpublished data) and was the temperature used during primary generation of the SCs to the 4th passaged generation.  This information has been added to the text.  All prior work in in vitro culture began by Dr. Douglas McFarland with turkey satellite cells beginning in the 1980’s used 38°C for the standard culture conditions. Therefore, the 43°C of temperature, which is 5°C higher than the control temperature, is higher than turkey’s body temperature. The 33°C, is 5°C lower than the control in vitro culture temperature.

- Line 118: As described, the study does not appear to have any biological replication. Only technical replication by plating duplicate wells from the same pool of cells was conducted. If that is the case and there is some reason the study was not or cannot be biologically replicated, this does need to be explained and acknowledged as a limitation. Ways in which individual variation was captured in the data set should be provided (e.g. Were the satellite cells at least a pool from multiple individuals? Were they from one sex or mixed sex? Why was the experiment not able to be replicated in time or with multiple pools of cells?)

Cells were isolated and stored prior to this study, not freshly isolated. This allows us to utilize a common biological source for interrelated experiments while controlling for individual variation.  Satellite cells used in this study were previously isolated from the p. major muscle of 20 male turkeys per turkey line.  Only utilized males to avoid sex related differences.  Experimental replicates are derived from the same starting material and are thus technical replicates for culturing and sequencing.  This is now noted in the text.

- Some table titles (in paper and supplemental) and figure legends refer to “F-line” cells and not NCT cells when it is clear they are referring to cells used in this experiment, which I think are different from “F-line cells”. This needs to be corrected throughout. If I am misunderstanding the “F-line” is synonymous the “NCT”, the line should be consistently referred to.

We apologize for the confusion.  Yes, the F-line and NCT line are not synonymous and there were instances (in the supplemental tables) where F mistakenly was used in place of NCT. Having worked with these lines for many years (RBC2 and F) it is hard to not include "F-line" when discussing comparisons to RBC2. These mistakes have now been corrected.

- Most figures and tables (including supplemental) need additional information so they can stand-alone. For example, in Table 2 “R” and “N” are used as abbreviations for the lines and not defined and it is not immediately clearly by the layout of the table how the “Unique Genes by Group” column is set up. There are also frequent abbreviations used in tables or figures that are not defined (e.g. FDR, FC, Diff, Pro, etc) and color coding in supplemental tables that is not described. While many of these things can be ultimately inferred, it is a better practice to have tables & figures fully able to stand-alone. All figures/tables need to be reviewed and changed as necessary.

Figures and Tables have been reviewed and updated accordingly.

-Supplemental Table 1:  No standard deviation is given as is stated in the title, so this needs to be added. It is not clear why the “overall mean” column is included, but this would be better titled as “mapped reads/sample”.

This table has been updated.  The error in the title (NCT vs F-line cells) has been corrected, standard deviations have been added for each group and the overall mean column has been renamed.

- Supplemental Table 2: The title does not describe information presented in the table and needs to be corrected. It might also be helpful to provide additional separate tabs for each comparison that only include differentially expressed genes that are sorted by fold difference.

This table was produced with the wrong captioned title.  This error has been corrected to reflect the information provided. We chose not to present this as separate tabs/tables as it is easier to directly compare the results for individual genes across comparisons in this format. As the table is provided in Excel format, interested readers can choose to individually sort on our data as they so desire.

- Lines 462 – 466 and lines 494 – 499 have identical text yet are describing different data. Assume this was a copy/paste oversight, but this needs to be corrected. What the numbers after each categories mean and where they were derived from needs to be provided, since they do not appear to come from Table 4.

This appears to have occurred during insertion of Table 4 into the Genes template.  In reviewing and checking information for this section, it was discovered that the GO Ontology database (DOI: 10.5281/zenodo.6399963) used for our analysis (released 2022-03-22) was recently updated (10.5281/zenodo.6799722 released 2022-07-01). Thus for completeness, we have repeated all of the PANTHER overrepresentation tests presented in this study using this new data release.  Application of this new release improved our analyses as it resulted in an increase in the number of turkey genes for which Gallus gallus homologs are identified.  Table 4 and text and  have been updated where appropriate.

- Lines 542 – 546, Supplementary Tables 3 & 4: It is unclear why this comparison was made, so a rationale for including it needs to be provided. It is acknowledged that analysis for each were done using different genome annotations, and both line and age are different so confound any ability to attribute differences in response between the groups to thermal challenge. The authors need to provide a stronger justification or remove this section.

These comparisons were made to examine the response of SCs of two different “selected” bird lines. The NCT birds are modern commercial birds. The F-line was a research line selected for body weight.  This line was used for several years as an example of growth selected birds.  As indicated in the Introduction to the paper, the comparisons contrast the responses of these cells and thus support the findings of earlier studies conducted on the F-line birds. This also provides a comparison between the cells derived from 7-wk and 1-wk birds.  Tables S3 and S4 have been updated.

- Lines 683 – 703: This section of the discussion seems entirely irrelevant to the data presented or the premise of the study as written. It is unclear why it is included, so either should be removed or a stronger connection made with data presented.

We believe that the results of our studies on the response of turkey SCs under thermal stress provides insight into muscle myopathies that affect poultry.  We have reworded this section to better tie these observation to our results.